

# Technical Note: Volume Transport Equations in Combined Sverdrup-Stommel-Munk Dynamics without Level of no Motion

Peter C. Chu
Naval Ocean Analysis and Prediction Laboratory, Department of Oceanography
Naval Postgraduate School, Monterey, CA 93943, USA

Correspondence to: Peter C. Chu (pcchu@nps.edu)

**Abstract.** The cornerstone theories of ocean dynamics proposed by Sverdrup (1947), Stommel

(1948), and Munk (1950) are based on the assumption of level of no motion. Such an assumption is

the same as the assumption of no meridional geostrophic transport. Ever since Sverdrup (1947)

however,   verification of the accuracy of the Sverdrup balance theory is based on the comparison

of the Sverdrup meridional transport with the meridional transport calculated directly from the

geostrophic currents based on hydrographic data. To overcome the mismatch between theory (no

meridional geostrophic transport in Sverdrup transport) and verification (comparison of Sverdrup

transport to meridional geostrophic transport), extended Sverdrup-Stommel-Munk transport

equations are derived in this note with replacing the level of no motion by the ocean bathymetry

and in consequence one forcing function (surface wind stress) in the classical transport equations

(with level of no motion assumption) is replaced by five forcing functions: density, surface wind

stress, bottom meridional current, bottom stresses due to vertical and horizontal viscosities. The

first two forcing functions (density and surface wind stress) are more than an order of magnitude

stronger than the other three forcing functions using the world ocean bathymetry, climatological

annual mean hydrographic and surface wind stress data. The extended Sverdrup volume transport

streamfunctions under wind forcing, density forcing, and combined wind and density forcing are

presented.



## 1. Introduction

The seminal papers by Sverdrup (1947), Stommel (1948), and Munk (1950) laid the foundation of

wind-driven ocean circulation. Sverdrup balance (SB) represents the meridional volume

transports employing only the local wind-stress in a linear dynamical framework (Wunsch 2011).

Stommel (1948) and Munk (1950) linear frictional ocean models are used to explain the existence

of intensive western boundary currents. Their theories were established on the base of an

assumption of level of no motion ($H$). What is the physics behind this assumption? What happens

if level of no motion does not exist?

Let the Cartesian coordinates be used with ($x$, $y$) the horizontal and $z$ the vertical

coordinates (upward positive) and ($\mathbf{i}$, $\mathbf{j}$, $\mathbf{k}$) the corresponding unit vectors, and let the horizontal and

vertical velocities be represented by $\mathbf{V} = (u, v, w)$. With low Rossby and Ekman numbers, steady

state momentum and continuity equations are given by


$$-f\rho v = -\frac{\partial p}{\partial x} + \rho A_z \frac{\partial^2 u}{\partial z^2} + \rho A_h \nabla^2 u , \tag{1a}$$

$$f\rho u = -\frac{\partial p}{\partial y} + \rho A_z \frac{\partial^2 v}{\partial z^2} + \rho A_h \nabla^2 v , \tag{1b}$$

$$\rho g = -\frac{\partial p}{\partial z} \tag{1c}$$

$$\frac{\partial(\rho u)}{\partial x} + \frac{\partial(\rho v)}{\partial y} + \frac{\partial(\rho w)}{\partial z} = 0 \tag{2}$$

where $f = 2\Omega \sin\varphi$, is the Coriolis parameter, $\Omega$ the Earth rotation rate, and $\varphi$ the latitude; $\rho$ is the

density; $p$ is the pressure; ($A_z$, $A_h$) are the vertical and horizontal eddy viscosities;

$\nabla = \mathbf{i}\partial/\partial x + \mathbf{j}\partial/\partial y$, is the horizontal gradient operator. Vertical integration of the horizontal



momentum equations (1a) and (1b) with respect to $z$ from a depth $z = -H(x, y)$ to the surface leads to

$$-A_h \nabla^2 M_x - f M_y = -\int_{-H}^{0} \frac{\partial p}{\partial x} dz + \tau_x - \tau_x^{(b)} - A_h \rho_0 Q_x \tag{3}$$


$$-A_h \nabla^2 M_y + f M_x = -\int_{-H}^{0} \frac{\partial p}{\partial y} dz + \tau_y - \tau_y^{(b)} - A_h \rho_0 Q_y \tag{4}$$

where

$$M_x = \int_{-H}^{0} \rho u \, dz, \quad M_y = \int_{-H}^{0} \rho v \, dz \tag{5}$$

are the zonal and meridional transports, which satisfy

$$\frac{\partial M_x}{\partial x} + \frac{\partial M_y}{\partial y} = 0 . \tag{6}$$

The vectors

$$(\tau_x, \tau_y) = \rho_0 A_z (\frac{\partial u}{\partial z}, \frac{\partial v}{\partial z}) \big|_{z=0} \tag{7a}$$

$$(\tau_x^{(b)}, \tau_y^{(b)}) = \rho_0 A_z (\frac{\partial u}{\partial z}, \frac{\partial v}{\partial z}) \big|_{z=-H(x,y)} = \begin{cases} C_D \rho_0 \sqrt{u_{-H}^2 + v_{-H}^2}\,(u_{-H}, v_{-H}), & \text{Drag law} \\ \\ R(M_x, M_y), & \text{Rayleigh friction} \end{cases} \tag{7b}$$

$$(Q_x, Q_y) = [(2\nabla u_{-H} \bullet \nabla H + u_{-H} \nabla^2 H), \ (2\nabla v_{-H} \bullet \nabla H + v_{-H} \nabla^2 H)], \tag{7c}$$

represent wind stress, bottom stress due to vertical eddy viscosity, and bottom stress due to

horizontal viscosity. Here, $(u_{-H}, v_{-H})$ are the current velocities at depth $z = -H$; $\rho_0$ (= 1028 kg/m$^3$) is the characteristic density.

If $z = -H$ is a level of no motion, $(u_{-H}, v_{-H}) = 0$ (also implying no stress at this level, i.e., $\boldsymbol{\tau}^{(b)}$ = 0 if the drag law is used) and the $P$ function in Sverdrup (1947)



$$\frac{\partial P}{\partial x} = \int_{-H(x,y)}^{0} \frac{\partial p}{\partial x} dz, \quad \frac{\partial P}{\partial y} = \int_{-H(x,y)}^{0} \frac{\partial p}{\partial y} dz, \tag{8}$$

exists.

Under the condition (8) (i.e., existence of the $P$ function) cross differentiation of (3) and (4) will make disappearance of the horizontal pressure gradient terms and give the classical Sverdrup balance (SB) if no horizontal viscosity ($A_h = 0$),

$$\beta M_y = \text{curl}(\boldsymbol{\tau}) \tag{9}$$

If $z = -H$ is not a level of no motion, the $P$ function does not exist. This is because

$$\frac{\partial}{\partial y}(\frac{\partial P}{\partial x}) = \frac{\partial p}{\partial x}\Big|_{-H(x,y)} \frac{\partial H}{\partial y} + \int_{-H(x,y)}^{0} \frac{\partial^2 p}{\partial x \partial y} dz,$$
$$\frac{\partial}{\partial x}(\frac{\partial P}{\partial y}) = \frac{\partial p}{\partial y}\Big|_{-H(x,y)} \frac{\partial H}{\partial x} + \int_{-H(x,y)}^{0} \frac{\partial^2 p}{\partial x \partial y} dz. \tag{10}$$

This leads to an impossible relationship

$$\frac{\partial}{\partial y}(\frac{\partial P}{\partial x}) \neq \frac{\partial}{\partial x}(\frac{\partial P}{\partial y}). \tag{11}$$

Cross differentiation of (3) and (4) without a level of no motion leads to a revised Sverdrup

transport equation,

$$\beta M_y = \left[ \frac{\partial}{\partial y} \int_{-H}^{0} \frac{\partial p}{\partial x} dz - \frac{\partial}{\partial x} \int_{-H}^{0} \frac{\partial p}{\partial y} dz \right] + \text{curl}(\boldsymbol{\tau}) - \text{curl}(\boldsymbol{\tau}^{(b)}) \tag{12}$$

Substitution of the geostrophic balance

$$f\rho u_g = -\frac{\partial p}{\partial y}, \quad f\rho v_g = \frac{\partial p}{\partial x} \tag{13}$$

into (12) gives

$$\beta M_y = \beta M_y^{(g)} + \text{curl}(\boldsymbol{\tau}) - \text{curl}(\boldsymbol{\tau}^{(b)}) \tag{14}$$

where


$$M_y^{(g)} = \int_{-H}^{0} \rho v_g \, dz = \frac{1}{f} \int_{-H}^{0} \frac{\partial p}{\partial x} \, dz \qquad (15)$$

Comparison between (9) and (14) leads to the fact that ***existence of a level of no motion is the same as vanish of meridional geostrophic transport in the system,***

$$M_y^{(g)} = 0. \qquad (16)$$

However, verification of the accuracy of the SB theory is based on the comparison of the Sverdrup meridional transport (i.e., the surface wind stress curl) with the meridional transport calculated directly from the geostrophic currents based on hydrographic data (i.e., the meridional geostrophic transport). The first was that of *Leetmaa et al*. [1977], followed by the studies of

*Wunsch* and *Roemmich* [1985], *Böning et al*. [1991], *Schmitz et al*. [1992], etc. Their results have shown that the Sverdrup meridional transport is generally consistent with the meridional transport calculated directly from the geostrophic currents based on hydrographic data in the northeastern subtropical North Atlantic Ocean, but is inconsistent with the geostrophic transports in the northwestern subtropical North Atlantic Ocean. *Meyers* [1980] discussed the meridional transport

of North Equatorial Countercurrent in the equatorial Pacific and found significant inconsistency with the Sverdrup theory. *Hautala et al*. [1994] estimated the meridional transport of the North Pacific subtropical gyre along 24°N and noted that the Sverdrup balance is not valid in the western subtropical Pacific Ocean.    Lately, *Wunsch* [2011] has evaluated the accuracy of the Sverdrup theory using an assimilated global ocean dataset.

A logical way to overcome such a mismatch between SB theory (no meridional geostrophic transport) and verification (comparison of surface wind stress curl to meridional geostrophic transport) is to remove the level of no motion and instead to use a known level. The ocean bottom [i.e., $z = -H(x, y)$] is a reasonable choice. Therefore, $H$ is referred to the ocean bottom depth here



after. Questions arise: How do the Sverdrup-Stommel-Munk equations change after $H$ is changed

from the level of no motion to the ocean bottom depth? What is new physics behind such a

change? This note will answer these questions. Following the same path as SB from (9) to (14),

several new volume transport equations (called extended Sverdrup-Stommel-Munk equations) have

been derived. The rest of the note is outlined as follows. Section 2 presents extended Sverdrup-

Stommel-Munk transport equations. Section 3 depicts the world ocean climatological annual mean

forcing functions. Section 4 shows the world ocean climatological annual mean density and wind

driven Sverdrup transport streamfunctions. Section 5 gives the summary.

## 2. Extended Sverdrup-Stommel-Munk Transport Equations

### 2.1. Geostrophic Currents under Boussinesq Approximation

With the Boussinesq approximation, vertical differentiation of (13) and use of hydrostatic balance

(1c) leads to the thermal wind relation,

$$\frac{\partial u_g}{\partial z} = \frac{g}{f \rho_0} \frac{\partial \rho}{\partial y}, \quad \frac{\partial v_g}{\partial z} = -\frac{g}{f \rho_0} \frac{\partial \rho}{\partial x}. \tag{17}$$

Vertical integration of (17) from the ocean bottom [$z = -H(x, y)$] to any depth $z$ leads to the

calculation of the geostrophic currents from the density $\rho$,

$$u_g = u_{-H} + \frac{g}{f \rho_0} \int_{-H}^{z} \frac{\partial \rho}{\partial y} dz', \tag{18}$$


$$v_g = v_{-H} - \frac{g}{f \rho_0} \int_{-H}^{z} \frac{\partial \rho}{\partial x} dz'. \tag{19}$$

### 2.2. Volume Transport Equations

Cross differentiation of (3) and (4) leads to the transport equation for the whole water column,

$$-A_h \nabla^4 \Psi + \beta \frac{\partial \Psi}{\partial x} = \beta V_{den} + \beta H v_{-H} + \frac{\operatorname{curl} \boldsymbol{\tau}}{\rho_0} - \frac{\operatorname{curl} \boldsymbol{\tau}^{(b)}}{\rho_0} - A_h \left( \frac{\partial Q_y}{\partial x} - \frac{\partial Q_x}{\partial y} \right) \tag{20}$$



where $\Psi$ is the volume transport streamfunction

$$M_x = -\rho_0 \frac{\partial \Psi}{\partial y}, \quad M_y = \rho_0 \frac{\partial \Psi}{\partial x}; \qquad (21)$$

$V_{den}$ is the meridional geostrophic transport

$$V_{den} = -\frac{g}{f \rho_0} \int\limits_{-H}^{0} \int\limits_{-H}^{z} \frac{\partial \rho}{\partial x} dz' dz. \qquad (22)$$

When the Rayleigh friction is used for the bottom stress (7b), Eq.(20) becomes the extended Munk-Stommel model,

$$-A_h \nabla^4 \Psi + R \nabla^2 \Psi + \beta \frac{\partial \Psi}{\partial x} = \beta V_{den} + \beta H v_{-H} + \frac{\text{curl } \boldsymbol{\tau}}{\rho_0} - A_h \left( \frac{\partial Q_y}{\partial x} - \frac{\partial Q_x}{\partial y} \right) \qquad (23)$$

When the Rayleigh friction is used for the bottom stress (7b) and horizontal viscosity vanishes ($A_h = 0$), Eq.(20) becomes the extended Stommel model,

$$R \nabla^2 \Psi + \beta \frac{\partial \Psi}{\partial x} = \beta V_{den} + \beta H v_{-H} + \frac{\text{curl } \boldsymbol{\tau}}{\rho_0}. \qquad (24)$$

When $A_h \neq 0$, and the drag law is used for the bottom stress due to vertical viscosity, Eq(20)

reduces to the extended Sverdrup transport equation,

$$\beta \frac{\partial \Psi}{\partial x} = \beta V_{den} + \beta H v_{-H} + \frac{\text{curl } \boldsymbol{\tau}}{\rho_0} - \frac{\text{curl } \boldsymbol{\tau}^{(b)}}{\rho_0} - A_h \left( \frac{\partial Q_y}{\partial x} - \frac{\partial Q_x}{\partial y} \right). \qquad (25)$$

## 3. Forcing Functions

Various transport equations (20), (23), (24), and (25) show that the volume transport is generated by density ($\beta V_{den}$), bottom meridional current ($\beta H v_{-H}$), wind stress curl [$(\text{curl } \boldsymbol{\tau})/\rho_0$], bottom

stress curl due to vertical viscosity [$(\text{curl } \boldsymbol{\tau}^{(b)})/\rho_0$] and horizontal viscosity [$A_h \left( \partial Q_y / \partial x - \partial Q_x / \partial y \right)$]. The density forcing function is further calculated by





$$\beta V_{den} = -\frac{g}{R_E \rho_0} \cot\phi \int_{-H}^{0}\int_{z}^{0} \frac{\partial\rho}{\partial x} dz' dz, \quad \beta = \frac{2\Omega}{R_E}\cos\phi, \quad \Omega = 2\pi/86400 \text{ s}^{-1}, \; R_E = 6370 \text{ km} \quad (26)$$

The last three forcing functions depend on the bottom current velocities ($u_{-H}$, $v_{-H}$) [see (7b), (7c) and (20)], horizontal diffusivity $A_h$, and the bottom drag coefficient $C_D$. The P-vector inverse

method (Chu 1995, 2000, Chu et al. 1998a, b) is used to determine ($u_{-H}$, $v_{-H}$) from hydrographic data. The horizontal diffusivity ($A_h$) is taken the value of $1.5\times10^3 \text{ m}^2\text{s}^{-1}$ by Smargrinsky parameterization and the bottom drag coefficient $C_D$ is set as 0.0025 (see Chu and Fan, 2007).

In this study, the five forcing functions are calculated from the three global datasets: (1) climatological annual mean hydrographic data downloaded from the NOAA 's World Ocean Atlas

2013 at the website: http://www.nodc.noaa.gov/OC5/woa13/ for computing ($U_{den}$, $V_{den}$) [see (17)], (2) the NOAA 's ETOPO5 from the website: http://www.ngdc.noaa.gov/mgg/global/etopo5.HTML for bottom topography $H(x, y)$, and (3) the Comprehensive Ocean-Atmosphere Data Set (COADS) at http://iridl.ldeo.columbia.edu/SOURCES/.DASILVA/.SMD94/.climatology/ for computing climatological annual mean surface wind stress ($\tau_x, \tau_y$).

The climatological annual mean density forcing function $\beta V_{den}$ (Fig. 1) is much stronger than the wind forcing function (curl $\boldsymbol{\tau}$)/$\rho_0$ (Fig. 2). The root-mean-square (RMS) is $2.81\times10^{-10}$ m/s$^2$ for $\beta V_{den}$ and $6.16\times10^{-11}$ m/s$^2$ for (curl $\boldsymbol{\tau}$)/$\rho_0$. The other three forcing functions are much smaller. Fig 3 shows the bottom stress forcing function due to horizontal viscosity $[A_h(\partial Q_y/\partial x - \partial Q_x/\partial y)]$ with the RMS of $3.32\times10^{-12}$ m/s$^2$, which is almost two-orders of

magnitude smaller than the density forcing. The computation shows the following relationship,

$$\{[\text{RMS}(\beta H v_{-H}), \text{RMS}[\text{curl }\boldsymbol{\tau}^{(b)}/\rho_0], \text{RMS}[A_h(\partial Q_y/\partial x - \partial Q_x/\partial y)]\}$$
$$\ll \text{RMS}[\text{curl }\boldsymbol{\tau}/\rho_0] < \text{RMS}[\beta V_{den}]$$
(27)





Besides, the correlation coefficient between $\beta V_{den}$ and $(\mathrm{curl}\,\boldsymbol{\tau}^{(b)})/\rho_0$ is 0.0699, which implies the independence between the meridional geostrophic transport for the whole water column and the surface wind tress curl.

## 4. Density and Wind Driven Sverdrup Transport Streamfunctions


The extended Sverdrup Transport equation due to density and wind forcing [vanish of other three forcing functions in (25)] is given by

$$\beta \frac{\partial \Psi}{\partial x} = \beta V_{den} + \frac{\mathrm{curl}\,\boldsymbol{\tau}}{\rho_0}. \qquad (28)$$

If wind forcing vanishes, the density driven Sverdrup transport streamfunction is calculated by


$$\frac{\partial \Psi_{den}}{\partial x} = V_{den} \qquad (29)$$

If density forcing vanishes, the wind driven Sverdrup transport streamfunction is determined by

$$\beta \frac{\partial \Psi_w}{\partial x} = \frac{\mathrm{curl}\,\boldsymbol{\tau}}{\rho_0} \qquad (30)$$

which is the classical Sverdrup dynamics but the volume transport is for the whole water column rather than above a level of no motion. It is noted that in calculating the density forcing function

$\beta V_{den}$ in (28) and (29), the latitude $\phi$ is set as 15°N for the zonal region of 0°-15°N, and as 15°S for the zonal region of 0°-15°S [see (26)]. With the climatological annual mean density and wind forcing functions calculated in Scetion 3, the three Sverdrup transport euqations (28)-(30) are solved using the traditional method with $\Psi = 0$ at the east boudary and integrating westward to get the climatological annual mean density driven Sverdrup transport $\Psi_{den}$ (Fig. 4), wind driven

Sverdrup transport $\Psi_w$ (Fig. 5), and density-wind driven Sverdrup transport $\Psi$ (Fig. 6).

Since the purpose of this note is to present the extended transport equations after removing level of no-motion rather than to simulate the volume transports, only major features on ($\Psi_{den}$, $\Psi_w$,



Ψ) are discussed. Although the density and wind forcing functions are very different (Fig. 1 and Fig. 2) with low correlation coefficient (0.0699), the three Sverdrup transport streamfunctions

($\Psi_{den}$, $\Psi_w$, $\Psi$) show similar patterns in both hemispheres, i.e., subpolar gyre, subtropical gyre, equatorial current, and equatorial counter current. The volume transport driven by density ($\Psi_{den}$) is stronger than driven by wind ($\Psi_w$). The correlation coefficient is 0.185 between $\Psi_{den}$ and $\Psi_w$, 0.716 between $\Psi_{den}$ and $\Psi$, and 0.698 between $\Psi_w$ and $\Psi$.

## 5. Summary

The seminal theories by Sverdrup (1947), Stommel (1948), and Munk (1950) are established on the assumption of level of no motion. This note shows that this assumption is equivalent to the assumption of no meridional geostrophic transport. To remove the level of no motion and instead to use bottom topography, extended Sverdrup-Stommel-Munk transport equations are derived in this note with adding four more forcing functions in addition to the surface wind stress: density,

bottom meridional current, bottom stresses due to vertical and horizontal viscosities. The density and wind forcing functions are dominant using the world ocean bathymetry, climatological annual mean hydrographic and surface wind stress data. The density and wind forcing functions are independent with very low correlation coefficient (0.0699). However, the Sverdrup transport streamfunctions under density, wind, and both forcing show similar patterns in both hemispheres,

i.e., subpolar gyre, subtropical gyre, equatorial current, and equatorial counter current. The correlation coefficient is 0.185 between density and wind forced Sverdrup transports; 0.716 between density and density-wind forced Sverdrup transports; and 0.698 between wind and density-wind forced Sverdrup transports.

## Acknowledgments



The author thanks Mr. Chenwu Fan's outstanding efforts on computational assistance, NOAA

National Centers for Environmental Information (NCEI) for World Ocean Atlas 2013, bathymetry

data, and Atlas of Surface Marine Data.

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



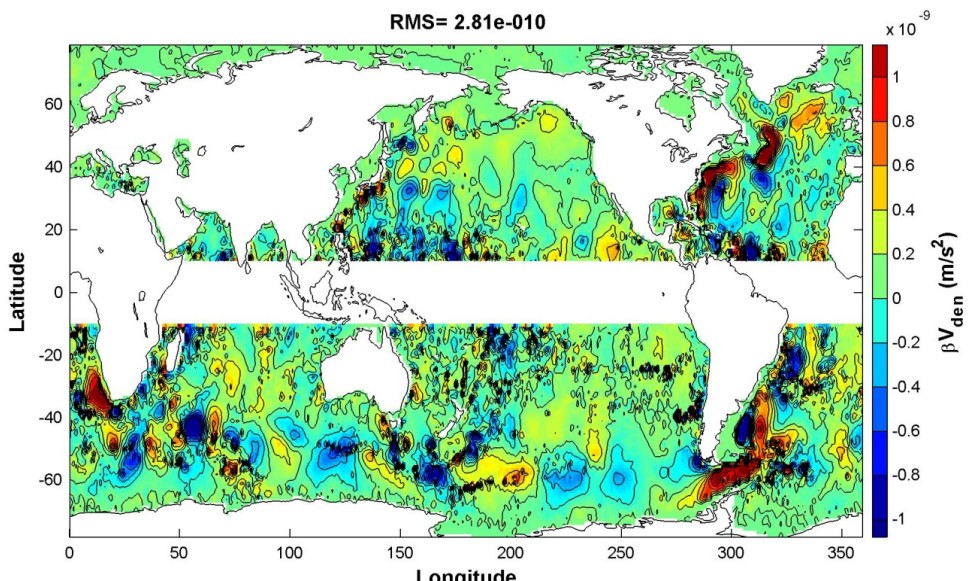

Fig. 1. Climatological annual mean density forcing $\beta V_{den}$ (unit: m/s$^2$) calculated from the NOAA/NCEI WOA13 data.


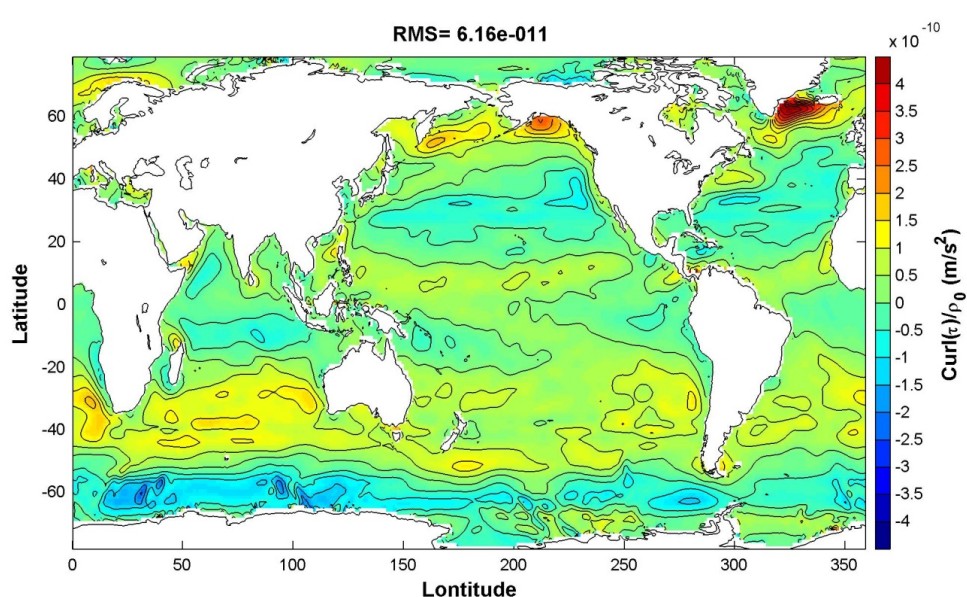

Fig. 2. . Climatological annual mean surface wind forcing $[(\text{curl } \boldsymbol{\tau})/\rho_0]$ (unit: m/s$^2$) calculated using the COADS data



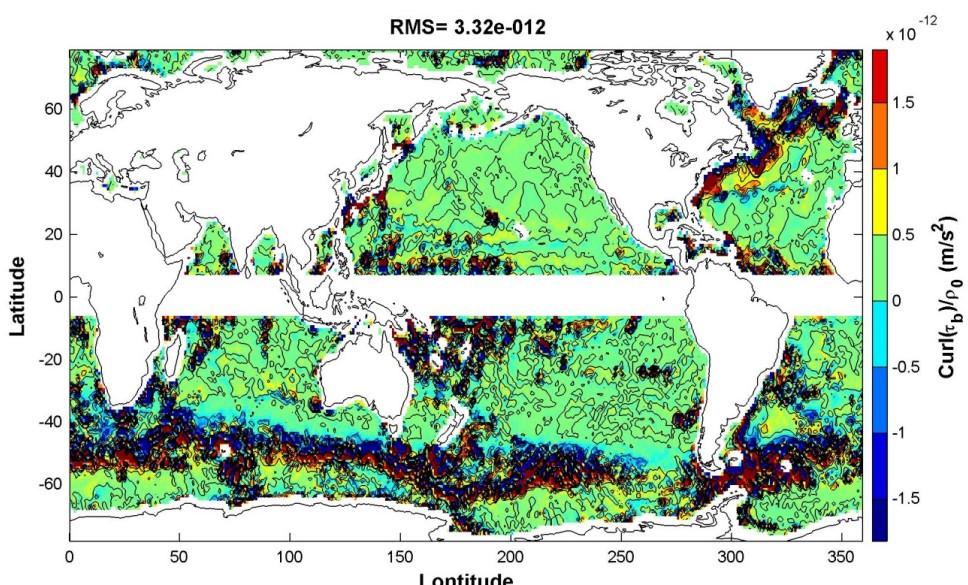

Fig. 3. Climatological annual mean bottom stress forcing due to horizontal viscosity
$[\, A_h \left( \partial Q_y / \partial x - \partial Q_x / \partial y \right) \,]$ (unit: m/s²) calculated from the NOAA/NCEI WOA13 data using the P-vector method and NOAA ETOPO5 data.

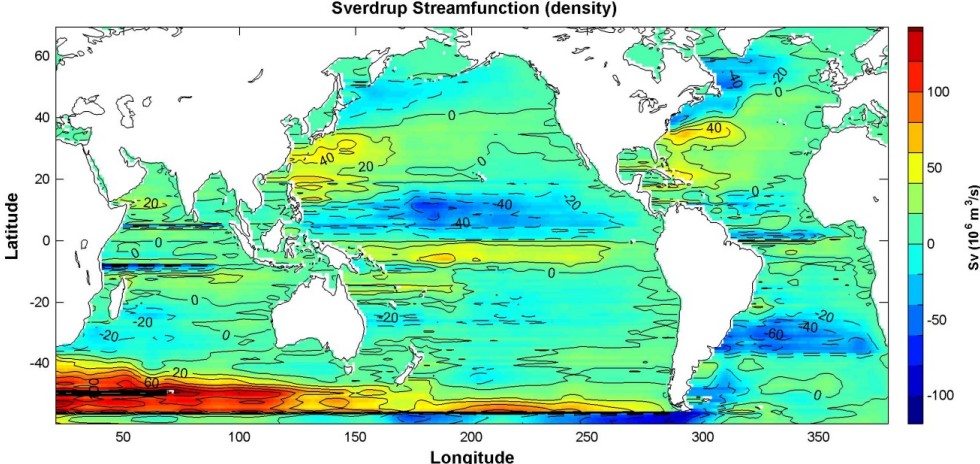

Fig. 4. Climatological annual mean density driven Sverdrup transport streamfunction (unit: S$_V$ =
$10^6$ m³/s). It is noted that in calculating the density forcing function $\beta V_{den}$,  the latitude $\phi$ is set as 15°N for the zonal region of 0°-15°N, and as 15°S for the zonal region of 0°-15°S.





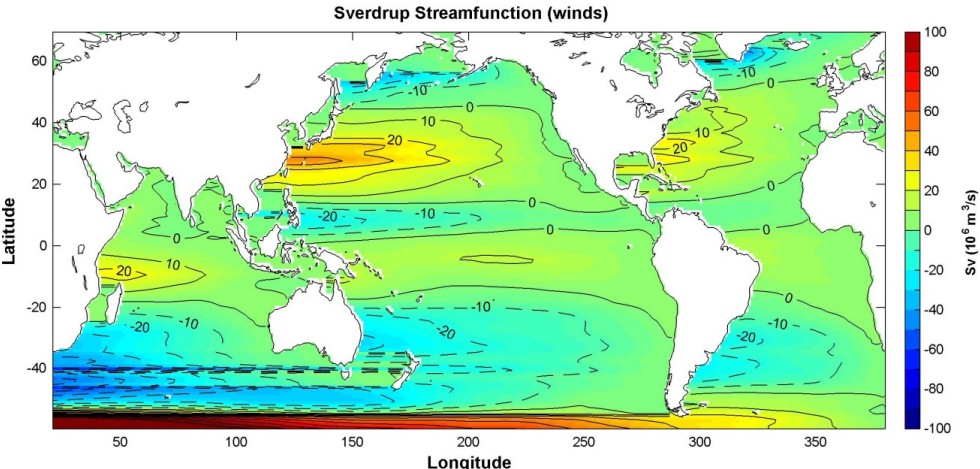

Fig.5. Climatological annual mean wind driven Sverdrup transport streamfunction (unit: $S_V = 10^6$ m$^3$/s).

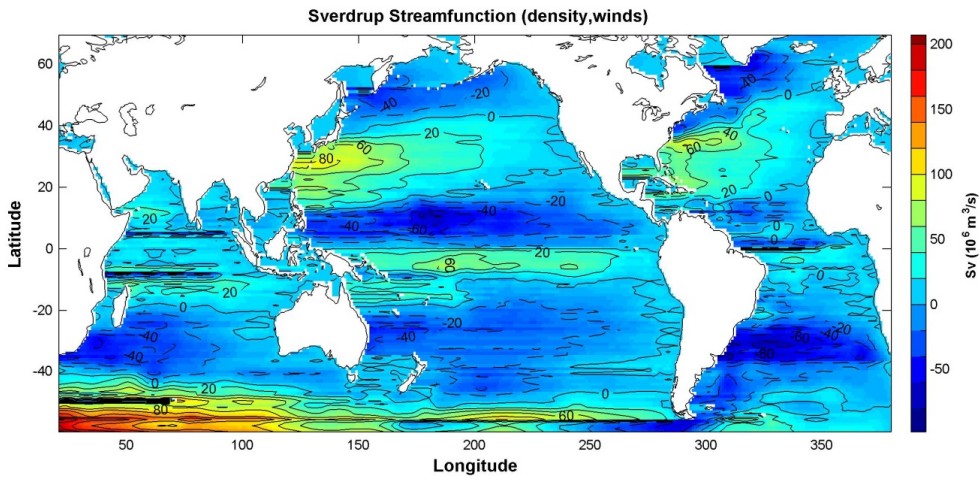

Fig. 6. Climatological annual mean density and wind driven Sverdrup transport streamfunction (unit: $S_V = 10^6$ m$^3$/s). It is noted that in calculating the density forcing function $\beta V_{den}$, the latitude $\phi$ is set as 15°N for the zonal region of 0°-15°N, and as 15°S for the zonal region of 0°-15°S.