# Peer review of "Technical Note: Volume Transport Equations in Combined Sverdrup-Stommel-Munk Dynamics without Level of no Motion"

_Ocean Science, 2016_

## Referee Comment (RC1) · Anonymous Referee #1 · 21 Nov 2016

The author reexamines Sverdrup dynamics and concludes that classic Sverdrup theory be replaced by extended equations including the effects of bottom topography, friction and stratification, in addition to wind forcing. Evaluations of the various terms show wind and density are the most important.

I'm afraid I cannot recommend publication of this paper. First, much is made of the result that geostrophic transport vanishes if there is a level of no motion. This is not true. The simplest counter-example is the 1.5 layer ventilated thermocline where the lower layer is assumed at rest, and yet the interior flow maintains a net meridional flow via Sverdrup dynamics. Eq (14) is missing a term proportional to the integrated divergence of the geostrophic flow which, without a level of no motion, is a term involving

the flow at depth -H. The only term retained in (14) is beta times meridional transport. Comparing then to (9) leads to the conclusion on page 5 about the vanishing of the geostrophic transport, which I disagree with.

The extended equation, including friction and bottom flow (20), is also missing this term. Wunsch and Roemmich show this quantity, i.e. the bottom flow in the presence of topography, is quite large. In spite of the statement that all other terms are shown to be small relative to wind and density, I could not find a plot of beta H V(-H). There seems to be a problem with (27) as it stands. Finally, the demonstration of (6) requires that the surface (-H) be a material surface and I didn't see a statement to that effect.

There are several grammatical errors and typographical errors in the manuscript.

In view of these issues, I can not endorse publication.

Interactive posting of a review is also a somewhat uncomfortable way to deliver such an opinion.

---

## Referee Comment (RC2) · Anonymous Referee #2 · 22 Nov 2016

This work has a number of fundamental flaws that prevent me from recommending it for publication. There are too many to connect all of the consequences to the final result, so I will just list them in the order they appear in the document.

1. The manuscript begins by stating small Rossby and Ekman numbers, but then retains the frictional terms and drops the acceleration and inertial terms. Thus, it is not a formally correct asymptotic limit, which would require boundary layers at the top and bottom where frictional terms are not negligible but a geostrophic interior flow.

2. It is unclear exactly how much density variation is to be preserved. (2) is a steady

state version of the compressible equations, but later in (3-7) $\rho_0$ is used in the fluxes and stresses without comment, which implies that a Boussinesq approximation is intended. The inconsistent treatment of density variations leads to mistakes in properly arriving at the JEBAR bottom boundary effects. An extensive literature on this topic, going back to Shtokman (1946), shows that the results presented here are not new (see Sarkisyan, 2006 and Sarkisyan and Sundermann, 2009). Even if these mistakes were corrected, the JEBAR approach can be misleading (Cane et al. 1998), and a vorticity budget needs to be carefully constructed so as to agree with its solution method–which is typically numerical nowadays (see Yeager, 2015 for a nice example).

3. Relatedly, the author proposes that the key depth $H$ can be made into a spatially-dependent function $H(x, y)$. Then, some of the terms required–which importantly include frictional boundary layers over a sloping bottom–are neglected in arriving at the form of the frictional terms in (3-4). (7c) does not fully handle the bottom boundary, because in those terms $\rho$ has been replaced with a constant, while in (5) they have not. In general, the author is careless with the Boussinesq approximation and the evaluation of the pressure gradients along the bottom boundary. (28) is not the vertically-integrated vorticity equation, because it is missing these interaction terms.

4. The surface is taken at $z = 0$ in (10), implying that a rigid lid approximation is being used, but a rigid lid approximation is inconsistent with (22), because the largest generator of near-surface geostrophic velocity is the gradient of sea surface height, or equivalently in the rigid lid approximation, the pressure boundary condition taken at $z = 0$. That is, (22) is *not* the meridional geostrophic transport.

5. It is stated that Stommel's (1948) solution relies on a level of no motion, which is incorrect. Stommel's original model is in fact a low Rossby number, uniform density, flat-bottom model with a spatially-varying sea surface height. Later papers revisiting Stommel's bottom drag balance added levels of no motion, or reduced gravity or equivalent barotropic reformulations to aid in re-interpretation (e.g., Fox-Kemper and Ferrari, 2009), but Stommel does not choose to assume a level of no motion.

6. (7b) is a terrible approximation over a sloping boundary in a baroclinic fluid. The along-boundary velocity, which experiences wave and frictional drag, is only lightly related to the depth-integrated mass transport. Indeed, in the Atlantic, beneath the Gulf Stream the bottom velocity is often flowing in the opposite direction of the surface current due to the deep western boundary current.

7. The primary reported results are (20, 23, 24, 25). All of these formulae are readily found (in more accurately derived versions) in the JEBAR literature, which also do not assume a flat bottom or a level of no motion.

8. The correct form of (30) is better interpreted as motion relative to f/H contours, see Holland (1967).

9. The overarching idea of this paper is that we do not have the machinery to evaluate the effects of forcing products without an assumption of a level of no motion, but we do this every day with GCMs. They produce solutions which are consistent with the vertically-integrated vorticity budget, which occasionally resembles the Sverdrup relation, and occasionally does not (see Yeager, 2015). Advanced parameterizations can be used for coupling to smaller scales, rather than inaccurate closures like (7b). Therefore, I am at a loss as to what the point here is, since the analytic work is not correct and if one is to use numerics then a better, more consistent solution is already in hand and published.

---

## Referee Comment (RC3) · Anonymous Referee #3 · 28 Nov 2016

In this article the author ostensibly extends the Sverdrup relation to include a "geostrophic" component and calculates "extended" transport streamfunctions for the real ocean using climatological data. The article is fundamentally flawed for the following reasons:

1) The premise of the article is fallacious because Sverdrup meridional transport is of course geostrophically balanced, including in the original gyre circulation theories of Stommel (1948) and Munk (1950). The author cites these articles in the first line of the abstract and the first line of the introduction, so it is perplexing that the manuscript could be so fundamentally confused on this point.

2) The author's additional "geostrophic" contribution to the Sverdrup transport results

from a mathematical error in equations (14) and (20), which incorrectly assume that the depth-integrated geostrophic transport is non-divergent.

3) The correct version of this additional contribution is the bottom pressure torque, whose contribution to the vortically-forced ocean circulation has already been discussed extensively in the JEBAR literature. The manuscript under review adds nothing to this literature.

There are various other issues with the work, ranging from the formulation of equations (1-2) to the many spelling and grammar errors in the manuscript, but I have chosen to omit them from this review because they will almost certainly be inconsequential.

The fundamental flaws listed above make the manuscript unpublishable in any form, so my recommendation is that the editor reject the manuscript.

---

## Editor Comment (EC1) · M. Hecht (Editor) · 9 Dec 2016

Dear Dr. Chu, based on the assessments of the referees, I have to conclude that there are fundamental issues that preclude further consideration of this work. I'm very sorry to say that we are rejecting your paper for publication.

You may either formally withdraw the paper at this point, or alternatively we will terminate the review process after the Discussion period ends later this month (I will be offline then, so it would be an additional week or so before this would be done).

I wish you well with your work.

Sincerely yours, –Matthew Hecht

---

## Author Comment (AC1) · 9 Dec 2016

Dear Dr. Hecht,

I would like to withdraw this manuscript.

Peter